# Residential Power Traces for Five Houses: The iHomeLab RAPT Dataset

**Patrick Huber \*,†, Melvin Ott †, Martin Friedli, Andreas Rumsch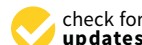 and Andrew Paice**

Engineering and Architecture, Lucerne University of Sciences and Arts, Technikumstrasse 21, 6048 Horw, Switzerland; melvin.ott@hotmail.com (M.O.); martin.friedli@hslu.ch (M.F.); andreas.rumsch@hslu.ch (A.R.); andrew.paice@hslu.ch (A.P.)

**\*** Correspondence: patrick.huber@hslu.ch
**†** These authors contributed equally to this work.

**Abstract:** Datasets with measurements of both solar electricity production and domestic electricity consumption separated into the major loads are interesting for research focussing on (i) local optimization of solar energy consumption and (ii) non-intrusive load monitoring. To this end, we publish the iHomeLab RAPT dataset consisting of electrical power traces from five houses in the greater Lucerne region in Switzerland spanning a period from 1.5 up to 3.5 years with a sampling frequency of five minutes. For each house, the electrical energy consumption of the aggregated household and specific appliances such as dishwasher, washing machine, tumble dryer, hot water boiler, or heating pump were metered. Additionally, the data includes electric production data from PV panels for all five houses, and battery power flow measurement data from two houses. Thermal metadata is also provided for the three houses with a heating pump.

**Keywords:** residential dataset; residential consumption; appliance power traces; PV production; non-intrusive load monitoring

## 1. Summary

Datasets with measurements of the domestic energy breakdown per appliance are of interest from different perspectives. Firstly, there is an ever-increasing amount of renewable electrical energy production and with that, a growing interest in solutions that can handle its stochastic nature such as demand response [1,2], optimized self-consumption [3] or smart energy trading [4]. Datasets that measure the consumption of aggregate households, as well as the (major) appliances, are required to base the development of such solutions on actual data. Secondly, datasets consisting of domestic electrical energy consumption with an appliance breakdown are essential for the non-Intrusive Load Monitoring (NILM) research, which aims at disaggregating the domestic energy consumption on device-level [5,6]. Finally, the data can also be useful for example in research in the domains of energy usage prediction, energy usage feedback systems, time-series data analysis and processing, and consumer behavior.

Our motivation in publishing this work is primarily to extend the list of available open datasets and therefore foster innovation in the corresponding fields. Our dataset consists of electrical power traces

from five houses in the greater Lucerne region in Switzerland spanning a period from 1.5 up to 3.5 years. For each house, the electrical energy consumption of the aggregated household and specific appliances such as dishwasher, washing machine, tumble dryer, hot water boiler, or heating pump were metered. The data includes additional electric production data from PV panels for all five houses, and power flow measurement data from a battery in the case of two houses. Three houses had a heating pump installed. For these houses, thermal metadata is also provided in order to enable corresponding simulations.

The dataset collection started in the framework of the project Wizard for the optimal management of Electrical Energy in a prosumer household (WizEE) [7] and is ongoing in the project Swiss Competence Center for Energy Research, Future Energy Efficient Buildings and Districts (SCCER FEEB&D) [8]. We, therefore, plan to release further data in the future. Based on the collected data, the authors investigated how well algorithms can predict the usage of domestic appliances solely based on historic electrical energy consumption data [9].

*Relation to Other Datasets*

A recent and comprehensive overview of datasets that have been used for load disaggregation work can be found e.g., in [10]. The dataset most similar to ours is named ECO [11] standing for Electricity Consumption and Occupancy dataset. It stems also from Switzerland and comprises a similar number of houses and duration. The main differences are the sampling frequency, 1 Hz in case of the ECO and 5 min for the presented dataset, and submetered devices. Our dataset includes data of appliances that consume a major fraction of the electrical household energy, i.e., heat pump and hot water boilers, but also photovoltaic production and household battery power flow measurements. The Dataport [12] database containing data from over 1000 residential homes is the most notable source of data that includes measurement of such heavy power consuming appliances. Available metering data stems however all from the US with quite distinct appliances and electrical installations compared to Switzerland. To our knowledge, datasets containing electrical power measurements with a similar breadth as Dataports are only available from the Irish Commission of Energy Regulation [13] and the 'Building Data Genome Project' [14]. The former contains however only 30 min aggregated residential data whereas the latter only hourly aggregated non-residential data.

## 2. Data Description

The dataset consists of data from five houses located in the greater Lucerne region. Each house is equipped with up to seven meters that measure each of the average consumed power per sampling period. Figure 1 contains a schematic overview of the installed (sub)metering and the corresponding wiring: The main meters of the houses are represented as thick red arrows. Thin red arrows indicate submetered appliances. Other electrical connections and corresponding meters are shown in blue/green. Sensor numbers are listed in Table 1. The table contains also a description of the sensors and their denomination in the dataset. The dataset is stored in the Hierarchical Data Format (HDF) version 5 [15]. If using the Python programming language, it is easily accessible with pandas [16,17] via the 'pandas.read_hdf(`path/to/file`)' function. All timestamps are given in Swiss local time, i.e., UTC+1:00h during winter half-year (November to March) and UTC+2:00h during the summer half-year (April to October).

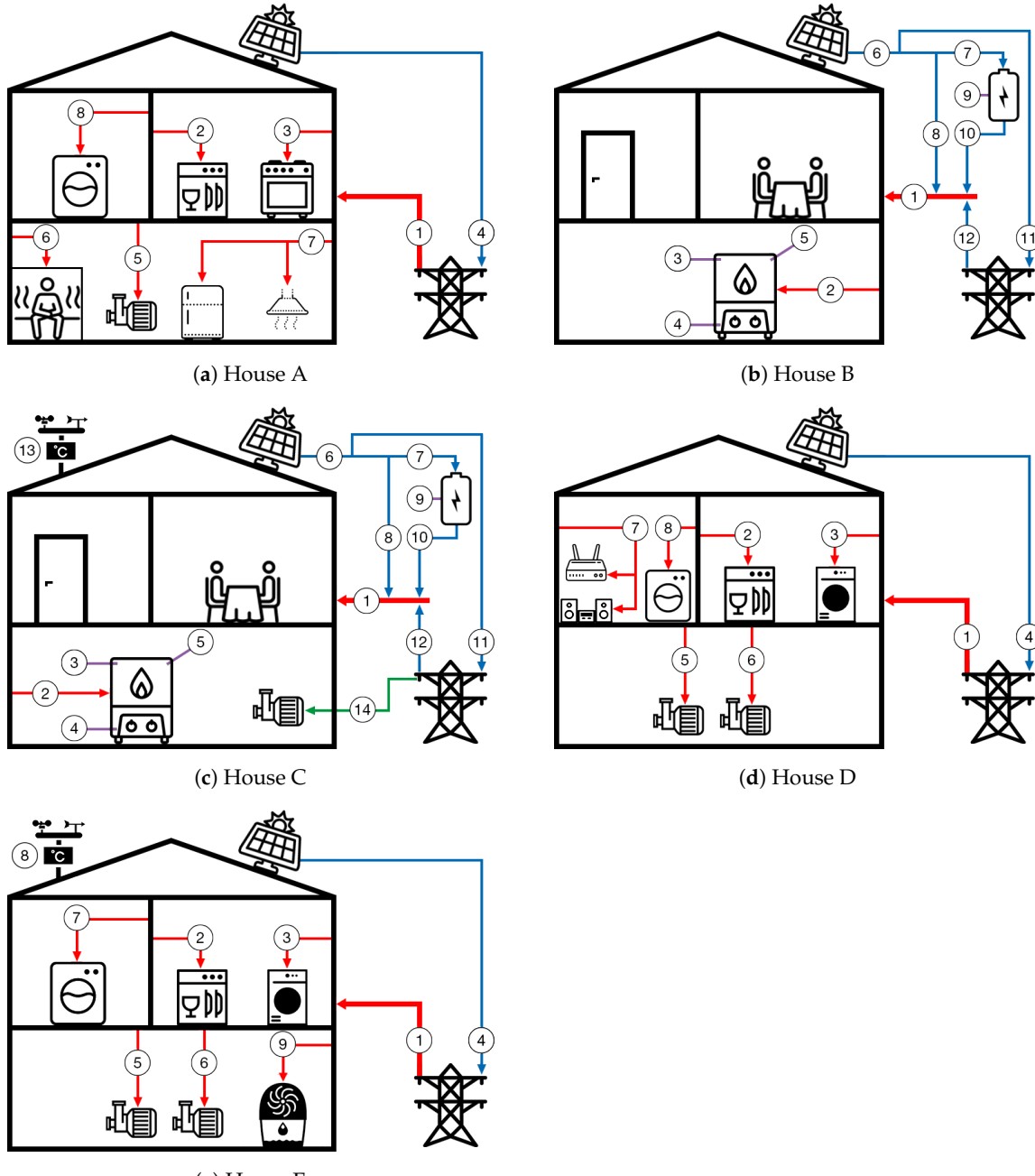

(**a**) House A

(**b**) House B

(**c**) House C

(**d**) House D

(**e**) House E

**Figure 1.** Schematic overview of installed meters and corresponding wiring: The main meters of the houses are represented as thick red arrows. Thin red arrows indicate submetered appliances. Other electrical connections and corresponding meters are shown in blue/green. The description for the sensor number is available in Table 1.

**Table 1.** List of sensors installed in the different houses. Numbers in column '#' refer to those in Figure 1. 'Sensor Name' gives the name of the sensor as used in the data. 'T$_{orig}$' indicates the original sampling rate. 'M-id' lists the meter identity as given in Table 5. Abbreviations: 'pc' stands for 'power consumption', 'PV' for 'photovoltaic', 'ed' for 'event driven'.

| # | Description | Sensor Name | Unit | T$_{orig}$ | M-id |
|---|---|---|---|---|---|
| | **House A** | | | | |
| 1 | total pc | A_total_cons_power | W | 120 s | a |
| 2 | pc of dishwasher | A_dishwasher_power | W | 120 s | b |
| 3 | pc of cooking, backing | A_stove_power | W | 120 s | c |
| 4 | PV power production | A_exp_power | W | 120 s | d, e [1] |
| 5 | pc of heat pump | A_hp_power | W | 120 s | f |
| 6 | pc of several devices: sauna, steam shower, outdoor lighting | A_sauna_power | W | 120 s | c |
| 7 | pc of several devices: fridge, comfort ventilation, central vacuum cleaner | A_additional_power | W | 120 s | b |
| 8 | pc of washing machine | A_washing_machine_power | W | 120 s | f |
| | **House B** | | | | |
| 1 | total pc | B_total_cons_power | W | 300 s | g |
| 2 | pc of boiler | B_boiler_power | W | 300 s | h |
| 3 | water boiler: temp. at the top | B_boilertemp_top | °C | 300 s | i |
| 4 | water boiler: temp. at the bottom | B_boilertemp_bottom | °C | 300 s | i |
| 5 | water boiler: on/off-status of heater 1, 2, 3 and thermostat [2] | B_boiler_heater_*, B_boiler_on_thermostat | – | 300 s ed | j |
| 6 | total PV power production | B_pv_prod_power | W | 300 s | g |
| 7 | from solar panel to battery | B_to_batt_power | W | 300 s | g |
| 8 | pc from solar panel directly | B_direct_cons_power | W | 300 s | g |
| 9 | battery charge state (100% = full) | B_batt_state | % | 300 s | g |
| 10 | pc from battery | B_from_batt_power | W | 300 s | g |
| 11 | from solar panel in grid | B_to_net_power | W | 300 s | g |
| 12 | pc from grid | B_from_net_power | W | 300 s | g |
| | **House C** | | | | |
| 1 | total pc | C_total_cons_power | W | 120 s [3] 300 s | a, g [3] |
| 2 | pc of boiler | C_boiler_power | W | 300 s | h |
| 3 | water boiler: temp. at the top | C_boilertemp_top | °C | 300 s | i |
| 4 | water boiler: temp. at the bottom | C_boilertemp_bottom | °C | 300 s | i |
| 5 | water boiler: on/off-status of heater 1, 2, 3, thermostat, relay, utility [2] | C_boiler_heater_*, C_boiler_on_* | – | 300 s ed | j |

**Table 1.** *Cont.*

| # | Description | Sensor Name | Unit | $T_{orig}$ | M-id |
|---|---|---|---|---|---|
| 6 | total PV power production | C_pv_prod_power | W | 300 s | g |
| 7 | from solar panel to battery | C_to_batt_power | W | 300 s | g |
| 8 | pc from solar panel directly | C_direct_cons_power | W | 300 s | g |
| 9 | battery charging state (100% = full) | C_batt_state | % | 300 s | g |
| 10 | pc from battery | C_from_batt_power | W | 300 s | g |
| 11 | from solar panel in grid | C_to_net_power | W | 300 s | g |
| 12 | pc from grid | C_from_net_power | W | 300 s | g |
| | metering of solar radiation | C_solarlog_radiation | $\frac{W}{m^2}$ | 120 s | k |
| 13 | environmental data logged by weather station, see Table 3 | C_weather_* | * | 1 h | l |
| 14 | pc of heat pump | C_hp_power | W | 120 s | a |
| 14 | ripple control signal for heat pump | C_hp_on_utility | - | ed | m |
| 15 | outdoor temperature next to sensor of heat pump | C_temperature_out | °C | 120 s | i |
| | | **House D** | | | |
| 1 | total pc | D_total_cons_power | W | 120 s | a |
| 2 | pc of dishwasher | D_dishwasher_power | W | 60 s | n |
| 3 | pc of tumble dryer | D_tumble_dryer_power | W | 60 s | n |
| 4 | PV power production | D_exp_power | W | 120 s | d |
| 5 | pc of heat pump | D_hp_power | W | 300 s | o |
| 6 | pc of rainwater pump | D_rainwater_power | W | 60 s | n |
| 7 | pc of HiFi system and router | D_audio_wlan_og_power | W | 60 s | n |
| 8 | pc of washing machine | D_washing_machine_power | W | 60 s | n |
| | | **House E** | | | |
| 1 | total pc | E_total_cons_power | W | 300 s | p |
| 2 | pc of dishwasher | E_dishwasher_power | W | 60 s | n |
| 3 | pc of tumble dryer | E_tumble_dryer_power | W | 60 s | n |
| 4 | PV power production | E_prod_power | W | 300 s | p |
| 5 | pc of heating gas pump | E_gasheating_pump_power | W | 60 s | n |
| 6 | pc of water pump of solar thermal collectors | E_solarheating_pump_power | W | 60 s | n |
| 7 | pc of washing machine | E_washing_machine_power | W | 60 s | n |
| 8 | environmental data logged by weather station, see Table 3 | E_weather_* | * | 1 h | l |
| 9 | pc of dehumidifier | E_dehumidifier_power | W | 60 s | n |

[1] Meter d was replaced with e on 2018-4-16.
[2] Please refer to Section 4.4 for the meaning of these sensors.
[3] Meter a ($T_{orig}$ = 120 s) removed on the 2017-09-26. Meter g ($T_{orig}$ = 300 s) installed on 2017-11-01.

## 2.1. Electrical Power Data

Electrical power measurements are available in a raw and processed version. The raw sensor measurements are provided one HDF-file per sensor combined in one folder per house. The processed data is joined in one HDF-file per house as summarized in Table 2. The table contains also the logged time

span for each house. The HDF-files consist of tables where each tables' column corresponds to one sensor reading.

**Table 2.** Summary of the presented dataset: Name of houses, corresponding Hierarchical Data Format (HDF)-files, sampling rate, available period and 'short name' of the relevant weather station of MetoSwiss for each house, see Section 2.2 for details.

| Houses | File Name | Sampling Rate | Time Range | Weather Station |
|---|---|---|---|---|
| house A | datasets/dfA_300s.hdf | 5 min | 2017-04-01–2018-10-30 | LUZ |
| house B | datasets/dfB_300s.hdf | 5 min | 2017-03-01–2019-07-31 | MMSTA |
| house C | datasets/dfC_300s.hdf datasets/dfC_3600s.hdf | 5 min 1 h (weather) | 2015-11-30–2019-07-31 | EGO |
| house D | datasets/dfD_300s.hdf | 5 min | 2016-04-23–2019-07-31 | LUZ |
| house E | datasets/dfE_300s.hdf datasets/dfE_3600s.hdf | 5 min 1 h (weather) | 2016-12-01–2019-07-31 | LUZ |

For the processed power data, we down-sampled some of the measured sensor data to have a consistent sampling interval of 5 min across all sensors of the five houses. This was mostly done for sensors of houses A, D and E. The original sampling rate of each sensor is given in column $T_{orig}$ of Table 1.

## 2.2. Weather Data

Houses C and E were equipped with a local weather station. The corresponding sensor descriptions and names are listed in Table 3. Sampling was done on an hourly basis. The available HDF-files are given in Table 2. As house D is in close proximity to E, corresponding outdoor weather data can be used for both.

**Table 3.** Description of the data recorded by the 'My Weatherbox' in houses C and E. The technical specification of the weather station can be found in Table 5, sensor l.

| Description | Sensor Name | Unit | $T_{orig}$ |
|---|---|---|---|
| | **House C** | | |
| indoor humidity | C_weather_humidity_in | RH | 3600 s |
| outdoor humidity | C_weather_humidity_out | RH | 3600 s |
| outdoor pressure | C_weather_pressure | mbar | 3600 s |
| indoor temperature | C_weather_temperature_in | °C | 3600 s |
| outdoor temperature | C_weather_temperature_out | °C | 3600 s |
| | **House E** | | |
| indoor humidity | E_weather_humidity_in | RH | 3600 s |
| indoor humidity in cellar | E_weather_humidity_cellar | RH | 3600 s |
| outdoor humidity | E_weather_humidity_out | RH | 3600 s |
| outdoor pressure | E_weather_pressure | mbar | 3600 s |
| indoor temperature | E_weather_temperature_in | °C | 3600 s |
| indoor temperature in cellar | E_weather_temperature_cellar | °C | 3600 s |
| outdoor temperature | E_weather_temperature_out | °C | 3600 s |

For all houses, hourly averaged weather data can be obtained from the Swiss Federal Office of Meteorology and Climatology, MeteoSwiss, upon registration at https://gate.meteoswiss.ch/idaweb/login.do. This service is free of charge for educational and research purposes. The 'short names' of the relevant weather stations for each house are listed in Table 2.

## 2.3. Thermal Metadata

Houses A, C and D include the electrical consumption of the heat pump. To make the dataset also useful for research questions involving the thermal aspects of the heated buildings, relevant thermal characteristics of these three houses are provided in Table 4.

**Table 4.** Thermal metadata of houses A, C and D.

| | House A | House C | House D |
|---|---|---|---|
| | | **Building Information** | |
| Description | Semi-detached house with three floors: The lowest floor of the southern facade is partially built into the hillside. The window area on the main northern facade (three floors) is small. The window area on the main southern facade (two floors) is large. | Detached house with 4 floors: only the topmost floor is freestanding. The rest of the north-east (NE) facade and around 1/3 of south-east and north-west facades are embedded in the hillside. The main facade is oriented towards south-west (SW). The heated volume includes an additional self-contained flat. No electrical consumption (except heat pump) origins from that flat. | Detached house with 2 floors: only the upper floor is freestanding, the complete NE and SW facades of the lower floor are embedded in the hillside. The main facades are oriented towards NE and SW. |
| Inhabitants | three-person household | four-person household + one person in flat | five-person household |
| Heated space [m$^2$] | 146 | 295 | 218 |
| Building envelope | Construction year: 2005, one-shell brickwork, 16 cm Isolation | Construction year: 2005, two-shell brickwork: outer: 15 cm brick, insulation: 14 cm, flumroc.ch 'Dämmplatte 1', thermal conductivity: 0.035 W/mK, inner: 15 cm brick | Construction year: 2008, ground floor concrete, first floor 80 mm massive wood construction encased in 260–300 mm insulation. Energy balance according to SAI 380/1 (2007) 33 kWh/m$^2$ year. See also Section 4.4 |
| Windows | double glazing, heat transmission coefficient: 1.9 W/(m$^2$ K) | double glazing, heat transmission coefficient: 1.9 W/(m$^2$ K) | triple glazing |
| | | **Heat Pump Information** | |
| Comments | Heat pump has two compressors, one is used for domestic hot water (60 °C), the other for space heating (30–35 °C). | Heat pump is only used for space heating. Domestic hot water is generated in a conventional hot water boiler. | Heat pump has two compressors, one is used for domestic hot water (60 °C), the other for space heating (35–40 °C). |
| Heat pump specs | Manufacturer: KWT (acquired by Viessmann), Model: Swissline 28NHB, Type: Sole—Wasser | Manufacturer: Stiebel Eltron, Model: WPL 23, Type: air–water | Manufacturer: alpha-innotec, Model: SWC 60H, Type: sole–water |
| Emitter type | floor heating | floor heating | floor heating |
| Ripple controlled heat pump | No | Yes | No |

## 3. Methods

### 3.1. Measurement Equipment and Data Collection

Metering was done with the sensors listed in Table 5. In cases where the number of pulses from the S0 interfaces per capturing period was counted, the measuring method induces the uncertainty that a major fraction of the energy of the first pulse in a capturing period could actually have been consumed before that period. One consequence thereof is that even if the actual energy consumption of an appliance was constant, the measured signal could exhibit a variation $\leq$ (energy/pulse).

**Table 5.** List of employed measurement equipment. Indexes given in column meter id 'M-id' are used in Table 1. The S0 interface is defined in IEC standard 62053-31 and DIN standard 43 864.

| M-id | Device Name | Details |
|---|---|---|
| a | L+G ZMD120AP | Meter installed by utility. Readout of pulse output (S0): 1000 impulses/kWh. |
| b | MCI single-phase | Readout of pulse output (S0): 1000 impulses/kWh. |
| c | ABB OD4165 | Readout of pulse output (S0): 100 impulses/kWh. |
| d | L+G E350 | Readout of pulse output (S0): 1000 impulses/kWh. |
| e | Elster AS3000 | Readout of pulse output (S0): 1000 impulses/kWh. |
| f | Voltcraft DPM-314D | Readout of pulse output: 1000 impulses/kWh. |
| g | Fronius Symo Hybrid 5.0-3-S | Inverter from Fronius (www.fronius.com) with an integrated logging system to monitor the various power flows through the device. |
| h | RPi Boiler Control Unit | Custom made boiler control unit implemented on a Raspberry Pi that switches the three individual heating elements of the boiler. The `boiler_power` is *calculated* based on the configured value of the nominal consumption of the switched-on heating elements. The control logic is documented in Section 4.4, house B. |
| i | Temp. Sensor DS1820 | 1-wire bus temperature sensor DS1820 |
| j | Switch State Sensor | Sensor is connected to RPi Boiler Control unit. The information on the voltage drop over the switch is transformed in a binary state signal. |
| k | Tritec Spektron 320 | Soloar irradiation sensor from Tritec (www.tritec-energy.com). Measuring range 0-1500 W/m$^2$. Accuracy $\pm5\%$ annual mean. Considering an area and efficiency of a solar panel, the output of the sensor can be used to calculate the generated power of a corresponding installation. |
| l | TFA My Weatherbox | Weather station from TFA (www.tfa-dostmann.de). Temperature sensors - accuracy: $\pm1$ °C; humidity sensor–resolution: 1% RH, accuracy: $\pm5\%$ RH. |
| m | Ripple Control | Ripple control signal as obtained from the utility company. |
| n | myStrom WiFi Switch (CH) | Smart plug from myStrom (mystrom.ch). Measuring range 2–2300 W. Accuracy $\pm < 1\%$. |
| o | Hager EC311 | Readout of pulse output (S0): 10 impulses/kWh. |
| p | Solar-Log 1200 | Monitoring and energy management system from Solar-Log (www.solar-log.com). It includes options to export data. |

The data collection infrastructure is visualized in Figure 2: Measurements were collected in files directly by the myStrom WiFi Switches, the Fronius Symo Hybrid, and the Weatherbox. All other sensors were read-out by means of a Raspberry Pi (www.raspberrypi.org) and the collected data was then uploaded to an FTP server. All raw measurement files were finally imported into a database with custom scripts.

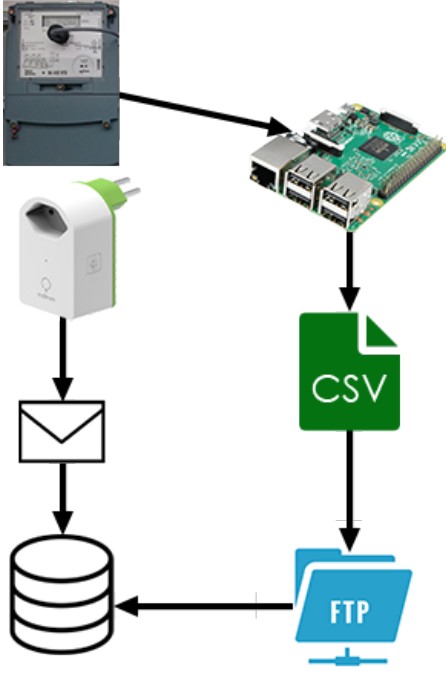

**Figure 2.** Overview of how the data was collected in general.

*3.2. Processing of Raw Data*

Multiple sensors in our dataset sampled with a higher frequency than once per five minutes and some sensors for the boilers of houses B and C were both event-driven and sampling at regular intervals. In order to provide an easily accessible dataset, we decided to publish a pre-processed dataset alongside the raw data. The corresponding code is available under https://github.com/ihomelab/RAPT-dataset. The pre-processing included the following steps:

- Round timestamps for the respective capture period, e.g., rounding '2018-10-01 18:05:08' to '2018-10-01 18:05:00'
- Convert electrical meter measurements to mean consumed power ([W]) per period.
- Linearly interpolate missing data for periods up to 15 min.
- Down-sample everything to a capture period of 5 min using the mean, except for the weather data, where an hourly resolution was kept.
- Calculate `boiler_power`: the `boiler_power` is calculated (not measured) in the RPi Boiler Control Unit, see sensor h in Table 5. Switch-on times are provided as sensors `C_boiler_heater_1\2\3`. As these sensors were also event-driven, the energy consumption was redistributed on the regular capture period of 5 min during pre-processing.

**4. Usage Notes**

*4.1. Reading the Dataset*

Reading the dataset in Python is straightforward. Using the Python package `pandas` [16], HDF files can be read with `read_hdf(...)`, as an example:

```
import pandas as pd
pathToHouseA = "datasets/dfA_300s.hdf"
```

```
dfA_300s = pd.read_hdf(pathToHouseA)
```

`dfA_300s` refers to a standard `pandas` DataFrame with the timestamp as index and available meters as columns.

*4.2. Missing Data*

Due to connection issues, reconstruction, and technical issues, the dataset contains missing data. It is stored as `NaN`. Figure 3 summarizes the corresponding information for the dataset (some weather sensors with similar missing data patterns were omitted). The different dates where logging started for the houses are easily observable. More detailed heat plots of the houses, showing the percentage of missing data per three days is provided alongside the data.

Using pandas DataFrames, the missing data can be easily extracted by `df[df['sensor_name'].isnull()].index` where `df` is the pandas DataFrame and `'sensor_name'` is the name of the respective sensor as a string. Additionally, we extracted the start and end dates of missing data for all the sensors and stored them in CSV files `missingData/<sensorName>.csv`. These files are provided alongside the data.

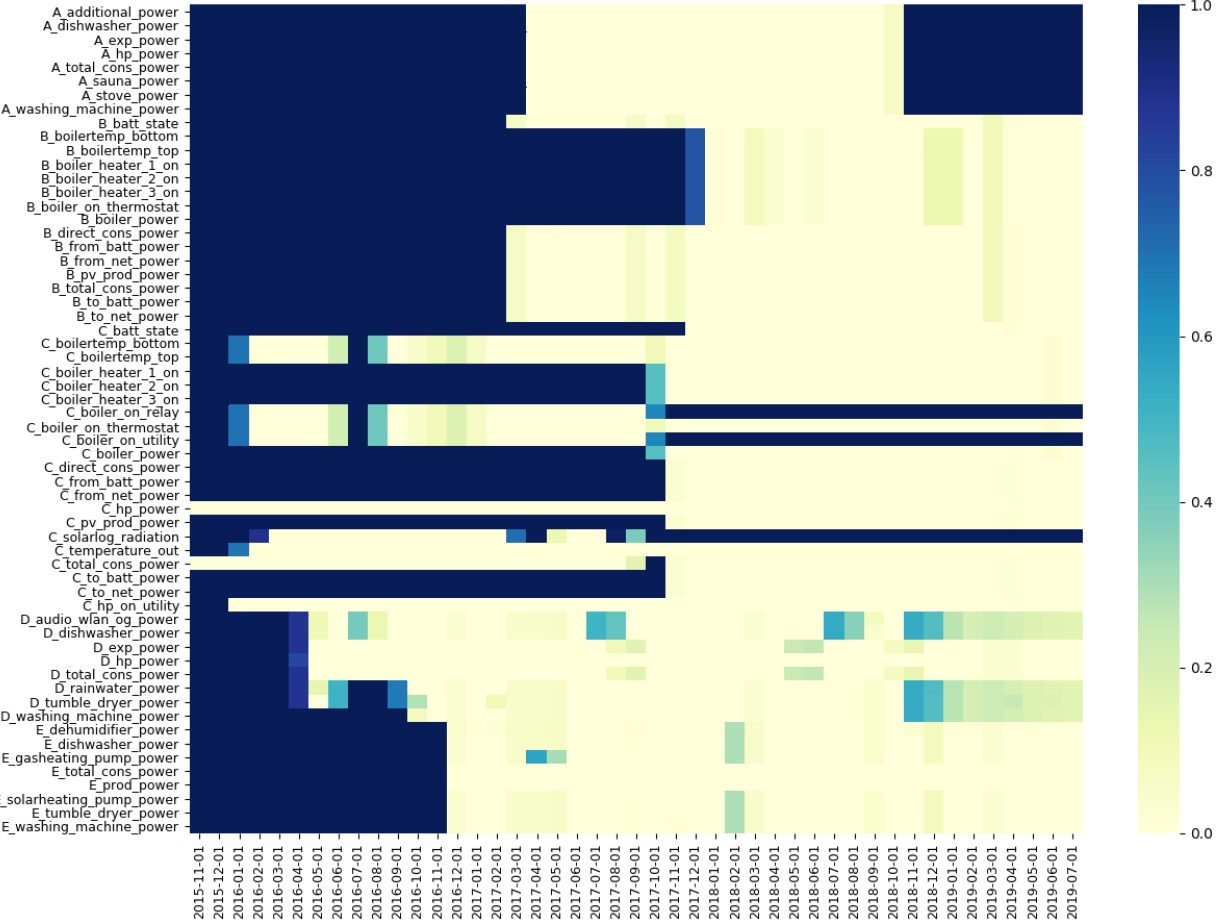

**Figure 3.** Heatplot of missing data for all houses (some weather sensors are omitted). The heatplot shows the percentage of missing data per month, i.e., 1 means no data (dark regions) and 0 means all data (brighter regions) is available.

*4.3. Known Issues*

A simple sanity check of the measured data consists of testing that the sum of all measured appliances must always be smaller or equal to the total consumed energy. In the case of houses B and C, there is a small portion of samples where this condition is not met, 0.07% and 0.28% respectively. These cases are mostly small deviations that can be attributed to the normalization of the data during the pre-processing steps, see Section 3.2.

*4.4. Particularities of the Installations and Corresponding Data*

House B

- Houses B has a solar panel, battery and a hot water boiler. The latter two are controlled by a custom controller, see sensor h in Table 5: If (`B_pv_prod_power - B_total_cons_power`) $> 600$ W, the `B_boiler_heater_*` are controlled according to the logic below:
- Daytime rules, i.e. high electrical tariff

  – **if** `boiler_on_thermostat == off` **then** `boiler_heater_* = off`
  – **if** `boiler_on_thermostat == on` **and** (`boilertemp_top` $< T_{min}$ **or** `boilertemp_bottom` $< T_{min}$) **then** `boiler_heater_* = on`
  – **if** (`boilertemp_top` $> T_{min}$ **and** `boilertemp_bottom` $> T_{min}$) **and** (`batt_state` $\leq$ 85%) **then** `boiler_heater_* = off`
  – **if** (`boiler_on_thermostat == 'on'`) **and** `batt_state` $> 85\%$ **and** `boilertemp_top`$< T_{max}$ **and** `boilertemp_bottom`$< T_{max}$ **then** `boiler_heater_* = on`

- Night-time rules, i.e. low electrical tariff

  – **if** `boiler_on_thermostat == on` **and** `boilertemp_top`$< T_{min\ night}$ **then** `boiler_heater_* = on`
  – **if** `boilertemp_top`$> T_{max\ night}$ **then** `boiler_heater_* = off`

  Here, $T_{min} = 40\,°C$, $T_{max} = 65\,°C$ and $T_{min\ night} = 35\,°C$, $T_{min\ night} = 45\,°C$. Depending on the current photovoltaic production, the three heating elements `boiler_heater_1/2/3` are *individually* switched on. Remaining power will, in all cases, be used to charge the battery to its maximum. Further excess power is only after that allowed to flow back into the electrical grid.
- Installed battery: 'Fronius Solar Battery 6.0' from Fronius (www.fronius.com): Usable capacity of the battery: 4800 Wh, nominal discharging and charging power 3200 W.

House C

- The family inhabiting house C was on an extended leave from 2016-06-24 until 2016-08-13 with corresponding low power consumption in that period.
- The heat pump of house C is directly connected to the electrical grid that means the corresponding power is not subsumed in `C_total_cons_power`. In exchange for a lower electrical tariff, the utility has the right to control switch-on times of the heat pump by means of ripple control. Before the 2016-04-01, the utility inhibited switch-on with minor exception from 11 to 12 o'clock, 15 to 18 o'clock and from 22 to 2 o'clock the following morning. Starting with the 2016-04-01, the utility is allowed to variably block the heat pump, where each blocked period has to be followed by a period of equal or longer duration within which the heat pump is allowed to pull power. The ripple control signal has been obtained from the utility and is available as sensor `C_hp_on_utility`, a value of 1 indicating the pump is allowed to pull power.
- Heating of the hot water boiler was controlled in the following ways:

- Before 2016-10-17: The boiler heating is turned on if the following signals are in the 'on' status: `C_boiler_on_utility` and `C_boiler_on_thermostat`. The former corresponds to the ripple control signal from the utility and the latter indicates if the water temperature in the boiler is below a certain threshold. The sensor `C_boiler_on_relais` is irrelevant for this period.
- Between 2016-10-17 and 2017-01-23: The boiler heating is turned on if in addition to `C_boiler_on_utility` and `C_boiler_on_thermostat`, the signal `C_boiler_on_relais` is also in status 'on'. The signal `C_boiler_on_relais` is switched according to the following logic: (i) 'on' if `C_boilertemp_top` < 42 °C (ii) 'off' if `C_boilertemp_top` > 48 °C (approximately every 14 days the threshold was manually set to 60 °C).
- Between 2017-01-23 and 2017-10-11: Due to recurrent difficulties with the more advanced control, the logic was switched back to the one used before 2016-10-17, see above.
- After 2017-10-15: After the retrofit, see below, battery charging and boiler heating is controlled by a custom controller, see sensor h in Table 5: If (`C_pv_prod_power` - `C_total_cons_power`) > 900 W, the `C_boiler_heater_*` are controlled according to the logic as described for house B with the following difference. In case the boiler is charged during the night, battery discharging is disallowed. The sensor `C_boiler_on_relais` is irrelevant for this period.

- 2017-09-26: Established electrical connection of new photovoltaic panels.
- 2017-11-29: Connected new battery to power system. The battery type is identical as for house B, see corresponding specifications.
- The installations of PV panels and battery had the following consequences on the electrical installation:

  - Before the PV installation, the solar irradiation was logged with a solar irradiation sensor Tritec Spektron 320, see Table 5. The corresponding sensor is called `C_solarlog_radiation` and is only available until the retrofit.
  - After the retrofit, electrical power consumption from the boiler was calculated by the RPi Boiler Control Unit, see sensor h in Table 5. This sensor was not available before. The power consumption level of the boiler before the retrofit did however not vary because heating elements were always jointly turned on. The consumed power can be indirectly deduced from the power steps in the aggregate signal.

House D

- The house is constructed as a "Zero-energy home"—Minergie-P, see Minergie (www.minergie.ch). That means that calculated heating energy per inhabited square meter and year amounts to 33 kWh.
- Solar panels have been disconnected between 2017-07-14 and 2017-10-18 because panels were exchanged. The area and exposition of the installed panels did not change but due to the improved efficiency, peak power changed from 8.14 kW to 10.45 kW.

House E

- A new dishwasher has been installed on the 2017-09-01.
- A new washing machine has been installed on the 2017-12-15
- The dehumidifier is located in the cellar. Its control logic ensures that the cellar is dehumidified at all times while maximizing the usage of excess solar power to this end. That means that every 10 min, the control logic checks if the relative humidity in the cellar exceeds an upper or falls below a lower limit and switches the dehumidifier on or off, respectively. Depending on the situation, different limits are applied. The limits are listed in Table 6.

**Table 6.** Situation dependent upper and lower limits employed in the control logic of the dehumidifier of house E.

| Situation | Lower Limit [RH] | Upper Limit [RH] |
|---|---|---|
| (B_pv_prod_power - B_total_cons_power) > 600 W (nominal power of dehumidifier) | 48 | 53 |
| high electrical tariff (starting at 6 am) | 58 | 63 |
| low electrical tariff (starting at 10pm) | 53 | 58 |

**Author Contributions:** P.H. and M.O. are the main authors: M.O. wrote the code to bring the raw data into a consistent format and the draft publication under supervision of P.H. who substantially refined text and content. M.F. installed all the meters and operated the data capturing systems. A.R. and A.P. assisted and advised throughout the project and publication writing. All authors have read and agreed to the published version of the manuscript.

**Acknowledgments:** This research project was financially supported by the Swiss Innovation Agency Innosuisse through the project 'Wizard for the optimal management of Electrical Energy in a prosumer household (WizEE)' project number 18064.2 PFEN-ES and is also part of the Swiss Competence Center for Energy Research SCCER FEEB&D. The authors want to also express their sincere thanks to the inhabitants of the houses involved in the datalogging. Their consent allowed our work and this publication. Icons in Figure 1 were obtained from www.icons8.de.

**Conflicts of Interest:** The authors declare no conflict of interest. The funding sponsors had no role in the design of the study; in the collection, analyses, or interpretation of data; in the writing of the manuscript, and in the decision to publish the results.

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
