# Peer review of "Residential Power Traces for Five Houses: The iHomeLab RAPT Dataset"

_data_

Round 1
Reviewer 1 Report
This paper provides the data set of residential power traces of five houses. It's interesting for the researchers and readers who focus on the home area energy consumption and optimization. So I suggest accepting this paper for publishing.
Author Response
The first reviewer commented on English spelling. The document has therefore been proof read and changes have been marked with a blue underline in the PDF file, generated with a new command "\spell" in the latex source file.
The PDF with the corresponding changes (irrespective of the second reviewers comments) has been attached.
We express our gratitude for the reviewers feedback,
Patrick Huber and co-authors.

Reviewer 2 Report
The article is very interesting. The measuring systems presented in it were described correctly and in detail. Defined open databases are clearly and legibly described. The data is original, sources well defined and have formats that may be available to other researchers. The final article summary is missing. They should answer such questions as: what benefits do the authors expect after completing this work? For what purposes can created databases be used? Did the authors achieve the intended effects?
Author Response
The second reviewer suggests to improve the following point:
"The final article summary is missing. They should answer such questions as: what benefits do the authors expect after completing this work? For what purposes can created databases be used? Did the authors achieve the intended effects?"
While we agree with the second reviewer that a concluding section is a very good practice in a research article, we think it is not necessary for the submitted paper, which is a Data Descriptor i.e. a publication that describes a dataset. We believe that the questions on possible benefits and the purpose raised by the second reviewer are already sufficiently covered in the section "Summary" of the submitted manuscript. A concluding remark would effectively be a repetition of what has already been stated in the section "Summary".
We still decided to strengthen the point on possible usages of the dataset in the section "Summary". The inserted sentence is marked in orange in the PDF version and with the command \reviewertwo in the latex source.
We express our gratitude for the reviewers feedback,
Patrick Huber and co-authors.
